# $C + 1$ Loss: Learn to Classify $C$ Classes of Interest and the Background Class Differentially

## Abstract

There is one kind of problem all around the classification area, where we want to classify $C + 1$ classes of samples, including $C$ semantically deterministic classes which we call Classes of Interest (CoIs) and the $(C + 1)^{\text{th}}$ semantically undeterministic class which we call background class. In spite of most classifier use softmax-based cross-entropy loss to supervise the training process without differentiating the background class from the CoIs, it is unreasonable as each of the CoIs has its inherent characteristics, but the background class dosen't. We figure out that the background class should be treated differently from the CoIs during training. Motivated by this, firstly we define the $C + 1$ classification problem. Then, we propose three properties that a good $C + 1$ classifier should have: separability, compactness and background margin. Based on these we define a uniform general $C + 1$ loss, composed of three parts, driving the $C + 1$ classifier to satisfy those properties. Finally, we instantialize a $C + 1$ loss and practice it in semantic segmentation, human parsing and object detection tasks. The proposed approach shows its superiority over the traditional cross-entropy loss.

## 1 Introduction

In machine learning, Softmax is one of the most widely used classifiers for classification, especially in CV tasks. During training, the Softmax classifier is often supervised by cross-entropy loss which treats each class without difference. However, there is a type of problem present all around the classification area, for which it is unreasonable to treat each class the same. We called this type of problem as $C+1$ classification. $C+1$ classification means classifying samples from $C$ semantically deterministic classes and the $(C + 1)^{\text{th}}$ semantically underterministic class. We can semantically uniquely define each of the $C$ classes by its inherent characteristics, so we can say they are semantically deterministic. Generally, we are interested in the $C$ classes, so we call them $C$ CoIs and one of them as a class of interest (CoI) hereafter. The $(C + 1)^{\text{th}}$ class includes all the other stuff beyond $C$ classes. Because it doesn't have uniform inherent characteristics and can't be described uniquely in semantics. That is why we say it is semantically undeterministic. In most cases, we regard things belonging to the $(C + 1)^{\text{th}}$ class as background, and we will call it background class hereinafter.

Based on the above description, we consider that it is reasonable to treat the $C$ CoIs and the background class differentially during training. On one hand, for the $C$ CoIs, it's reasonable to drive a $C + 1$ classifier to learn a compact and independent representation space for each of them because of their inherent characteristics. Then the $C + 1$ classifier can embed samples from each CoI into its own representation space. On the other hand, it's also reasonable to drive the $C + 1$ classifier to map any samples from the background class into somewhere in feature space far away from all representation spaces of the $C$ classes, considering that samples from background class doesn't have any inherent characteristics different from $C$ CoIs. For example in figure 1, if we are interested in cat, we can recognize a cat from an image at the first glance by the knowledge in the subconscious which uniquely defines cat. On the contrary, we can also recognize a not-cat instantly because it doesn't have any inherent characteristics of cat.

According to the inherent characteristics of $C + 1$ classification, we figure out that a $C + 1$ classifier will be preferable if it has a compact and independent representation space for each of the $C$ CoIs

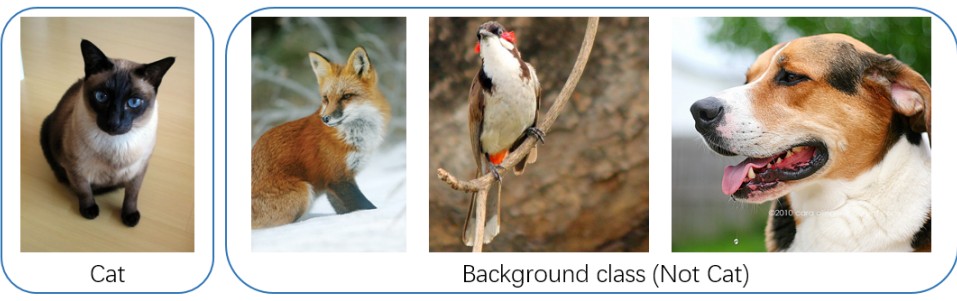

Figure 1: $C+1$ classification cases(the source images are from ImageNet Russakovsky et al. (2015))

in addition to its separability for all classes. This can guarantee it behaves well while encountering a sample which have different styles from the samples of corresponding CoI in the training set. Furthermore, it will be much better if there is large enough margin between the background class and the $C$ CoIs. This will make the classifier more robust and generalizable while encountering a sample from any new classes belonging to the super background class, especially those that never appear in the training set. Above on these, we conclude three properties which a good $C+1$ classifier should have—**separability**, **compactness** and **background margin**. Based on the three properties, we define a uniform general $C+1$ loss which includes three parts corresponding to the three properties, driving the $C+1$ classifier to satisfy those properties. At last, considering semantic segmentation and object detection are two of the most typical $C+1$ classification problems, and are widely used in AI systems, we instantialize a $C+1$ loss and practice it in semantic segmentation and object detection tasks, proving its superiority over the traditional cross-entropy loss.

Specifically, in semantic segmentation, Softmax is widely used to classify each pixel of an image to one label of a predefined class set. Typically the predefined class set includes $C$ CoIs and a background class. For instance, PASCAL VOC 2012 segmentation Everingham et al. (2010) contains 20 CoIs and a background class. The background class contains all other stuff. In object detection, Softmax is usually used to classify proposal boxes of an image to one of a predefined class set. For example, MS COCO Lin et al. (2014) contains 80 CoIs. A detector trained on it should classify all proposal boxes as one of the 80 CoIs or as background class. In these two typical tasks, the $C+1$ classifiers need to recognize all the CoIs, and classify a variety of other things as background. It's difficult for cross-entropy loss to drive the classifiers to learn well, which treats all samples without difference during training.

This paper contains three contributions summarized as follows:

1.We define the $C+1$ classification problem present all around the classification area.

2.We propose three properties that a good $C+1$ classifier should have, and define a uniform $C+1$ loss, which includes three parts driving the classifier to satisfy these properties.

3.We instantialize a $C+1$ loss consisting of three terms, and practice it on semantic segmentation and object detection tasks, proving its superiority over the popular cross-entropy loss.

## 2 RELATED WORK

### 2.1 SOFTMAX

Softmax is one of the most widely used classifier for a variety of pattern recognition tasks. Nowadays there are a plenty of variants for Softmax, such as L2-Softmax Ranjan et al. (2017), Large-margin Softmax Liu et al. (2017), Angular Softmax Liu et al. (2016), Normface Wang et al. (2017), AM-Softmax Wang et al. (2018a), CosFace Wang et al. (2018b) and ArcFace Deng et al. (2019). Large-margin Softmax is the first attempt to add parameter $m$ into the original Softmax to control the margin and the larger $m$ is, the larger the decision margin between the classes. Angular Softmax also known as SphereFace is an improvement to Large-margin Softmax with additional constraints on $W$ and $b$, introducing the hypersphere manifold which makes the features suitable for

open-set FR problem. L2-Softmax and Normface share similar ideas where L2-Softmax normalizes only the features and Normface normalizes both classifier weights and features and applys a scale parameter after that. The normaling and scaling steps push the learning progress focusing on optimizing angles among the classes, making the features not only separable but also discriminable. AM-Softmax(additive margin Softmax loss) and CosFace's works are inspired from SphereFace by moving the parameter $m$ that controls the margin from angular space to cosine space by addition. This also makes the implementation easier. ArcFace(additive angular margin loss) moves the parameter $m$ from scaling to addition to expand the optimization boundary. To sum up, thse variants of Softmax improve Softmax from these aspects: normalization of weights or features, margin in angular space or cosine space, setting of margin $m$. They use the parameter $m$ in different ways resulting in different decision boundary. However, they treat all classes equally during training as Softmax. And we propose to give some special care for the background class. So Softmax and its variants are not the best choice for $C + 1$ classification problem.

## 2.2 METRIC LEARNING

Metric learning aims to maximize inter-class variation and minimize the intra-class variations. Constrastive loss Chopra et al. (2005); Hadsell et al. (2006); Sun et al. (2014) drives the distances between positive pairs close to 0, and the distances between negative pairs to fall within an absolute range. Triplet loss and its variants Weinberger & Saul (2009); Hoffer & Ailon (2015); Wang et al. (2014); Schroff et al. (2015); Ding et al. (2015); Cheng et al. (2016); Oh Song et al. (2016); Sohn (2016) drive the relative distances between positive pairs and negative pairs to fall lower than a preset threshold. Center loss Wen et al. (2016) drives model to learn a center for features of each class and penalizes the distances between features and their corresponding class center. Metric learning also treats each class equally, just from a metric perspective. They could not take care of the particularity of background class and the inherent characteristics of every semantic class. In addition, it's not appropriate to learn a center for background class because it has no deterministic and unique definition.

## 2.3 OPEN-SET RECOGNITION

Open-set recognition Bendale & Boult (2016); Ge et al. (2017); Shu et al. (2017); Neal et al. (2018); Liu et al. (2019); Yoshihashi et al. (2019); Oza & Patel (2019); Chen et al. (2019a); Qian et al. (2019); Yu & Tao (2019); Sun et al. (2020) is the one closest to our proposed $C + 1$ classification, wherein there are no training samples for the background class. However, the classifier needs to detect new classes while inferring. The $C + 1$ classification deals with another type of problem, wherein there are both training samples for $C$ CoIs and the background class. In practice, it's impossible to contain all other semantic classes except the CoIs into the background class. Many samples of new semantic classes are likely yo be encountered and should be identified as background class while inferring. So we could call the $C + 1$ classification as semi-open-set recognition.

## 3 METHOD

### 3.1 PROBLEM DEFINITION

Firstly, we define some terminologies as follows.

**Semantic Class**: a set of samples which could be described uniquely and deterministically.

**CoIs**: a set of semantic classes that we are interested in and need to be recognized.

**Background Class**: all other classes beyond $C$ CoIs, including those semantic classes we are not interested and all other stuff.

$C + 1$ **Classification Problem**: categorize each item of a set into one of the $C + 1$ classes. Therein, $C+1$ classes comprise $C$ CoIs and the $(C+1)^{\text{th}}$ class of not interested, also called background class. The $C$ CoIs are deterministic and unique, but the background class is underterministic and includes all stuff beyond those belonging to the $C$ CoIs. The training set comprises many samples from each of the $C$ CoIs and diverse samples from the background class. Being semantically deterministic means we can give a deterministic and unique definition for each of the $C$ CoIs according to its

inherent characteristics. Being underministic means the background class has no deterministic and unique definition, because it lacks uniform inherent characteristics.

## 3.2 $C + 1$ Loss

For each of the $C$ CoIs, the $C + 1$ classifier should extract the inherent characteristics, and differentiate the background class from it. In order to achieve this, we consider that a $C + 1$ classifier should satisfy the following three key properties.

1.**Separability**: It should be able to separate all classes.

2.**Compactness**: The representation space of $C$ CoIs should be compact enough.

3.**Background margin**: There should be large enough space which we call background margin between the background class and each of the $C$ CoIs.

The property one guarantees that a $C + 1$ classifier has basic categorization ability for those samples that have similar distribution with the training set. The property two makes sure that the classifier has good generalizability for those samples belonging to the $C$ CoIs but with different distribution. And the property three gives the classifier good robustness and generalizability for samples of the background class, especially those whose styles are never present at the training set.

In order to drive a $C + 1$ classifier to learn to satisfy the three key properties, we argue that the $C + 1$ loss should include at least three parts: $L_{separability}$, $L_{compact}$ and $L_{background}$, corresponding to the three properties respectively.

$$L_{C+1} = L_{separability} + \alpha L_{compact} + \beta L_{background} \tag{1}$$

Herein, $L_{separability}$ drives the classifier learn to discriminate all classes without difference, $L_{compact}$ drives the classifier to try to grasp the inherent characteristics and to learn a compact representation space for each of the $C$ CoIs, and $L_{background}$ drives the classifier to differentiate the background class from the $C$ CoIs well.

In practice, we could adopt cross-entropy loss as an instantialized instance for $L_{separability}$ and center loss Wen et al. (2016) for $L_{compact}$. As for $L_{background}$, we design a novel loss according to the property three as follows.

$$L_{background} = \text{sign}\left(y_i = C + 1\right) \frac{1}{N} \sum_i \frac{1}{C} \sum_{k=1} |m_b - d\left(\boldsymbol{f}\left(\boldsymbol{x}_i\right), \boldsymbol{c}_k\right)|_+ \tag{2}$$

Herein, $\text{sign}\left(expression\right) = 1$ only when the $expression$ is true, otherwise $0$. $N$ is the number of background class sample. $m_b$ is the background margin between background class and each of the $C$ CoIs. $x_i$ is a sample, and $y_i$ is its class label index. $c_k$ is the center of $k^{\text{th}}$ CoI. $f(x)$ is a mapping function for extracting feature of sample $x$. $f(x)$ followed by softmax composes the $C + 1$ classifier. Without losing generality, $f(x)$ is a deep neural network.

As centers for center loss and the instantialized $L_{background}$, we define three types of center representation.

1.Represent each center of $C$ CoIs as a learnable weight vector.

2.Represent each center of $C$ CoIs as a moving average of sample features of the corresponding class.

3.Represent each center of $C$ CoIs as the classifier weight vector of each class from the logit FC layer.

Specifically the third center representation means that we share the center vector parameters with the softmax layer of the $C$ CoIs, and ignore the weight vector of background. We define a center for every CoI, but not the background class. For every CoI, there are a bunch of inherent characteristics which can uniquely and deterministically define it semantically, however there are no unique and deterministic inherent characteristics for the background class. However, we can calculate the loss

term $L_{compact}$ based on the distance between every sample of each CoI and its center. Furthermore, we can calculate the loss term $L_{background}$ based on the distance between every sample of background class and the center of each CoI.

## 3.3 APPLICATION

The $C + 1$ classifier described in the previous section can be applied to many tasks, including semantic segmentation, object detection, human pose estimation and any other classification problem which include the background class. In semantic segmentation, we label each foreground pixel with a semantic label, and all other pixels with background label. In object detection, we classify the area of object of interest as one of the CoIs, such as pedestrian, car and so on, and all other areas as background. In human pose estimation, we label the position of each human keypoint with proper semantic label and the other position with the background label. As for other cases, if the recognition problem needs to recognize certain number of semantically deterministic classes and the other stuff, the $C + 1$ classifier can also apply to it. In this paper, we just use some CV tasks as the experimental field, because they are the most common tasks in AI system. Besides the application scenarios presented above, it can also be applied to many other tasks, such as attribute recognition, web text classification, speech recognition and so on.

## 3.4 DISCUSSIONS

We think $C + 1$ classifier is rational because it has sufficient prerequisites. Because every CoI has semantically deterministic definition based on a set of inherent characteristics, it's rational and reachable to embed all samples of every CoI into an independent hypersphere. For example, the cat in the figure 1 has some type of fur, innate shape and contour, so we can define it uniquely and recognize it by the first glance. As for the background class, it has no deterministic definition, so there are no definite and uniform features for it. Then it's not rational to embed all samples of background class into an hypersphere. But it may be reasonable to map it to the outer space of all hyperspheres of the $C$ CoIs. Because the $C$ CoIs are separable from the background class based on the uniqueness of every CoI.

## 3.5 THEORETICAL ANALYSIS

After training, if the mapping function can embed all samples of the $C$ CoIs into their own hypersphere, it's almost impossible to map a novel sample belonging to the background class into any hypersphere of the $C$ CoIs, even though the classifier has no any knowledge about the novel sample. Assume that a novel sample belonging to the background class is new-brand and there are no similar style of samples present at the training set, then we can treat it as a random sample from the nature. From the perspective of statistics, the probability of recognizing it as the background class is close to 1. Formally, we assume that the full $d$-dimension feature representation space is a super ball with $R_L$ as the radius, and the sample features of each CoI are distributed inside a small super ball with $R$ as the radius. Then the probability of mapping the random sample to the outer space of the $C$ CoI super balls is

$$p\left(x_{\text{random}}\right) = 1 - \frac{C \cdot R^d}{R_L^d} \tag{3}$$

Therein, $R \ll R_L$ is a reasonable assumption, $C$ is a constant (number of CoIs), and $d$ is the dimension of feature space, usually very large, such as hundreds, even thousands. $C \cdot R^d$ is proxy for union of volumes of all CoI super balls, and $R_L^d$ for the full feature space. Then $\frac{C \cdot R^d}{R_L^d}$ is the ratio between union of volumes of all CoI super balls and the full full feature space, which represents the probability that the mapping function embeds a random sample of background class into any one of $C$ CoI super balls. In practice, because of the high dimension of feature space, which means $d$ is a large integer, such as $1024$, $p\left(x_{\text{random}}\right)$ is close to 1.

Table 1: Effectiveness validation experiment on PASCAL VOC

| Loss function | mIoU | mAcc | aAcc |
|---|---|---|---|
| $L_{separability}$ | 75.93 | 85.47 | 94.59 |
| $L_{separability} + L_{compact}$ | 76.25 | 86.1 | 94.61 |
| $L_{separability} + L_{compact} + L_{background}$ | **77.35** | **86.37** | **94.58** |

## 4 EXPERIMENT

We comprehensively evaluate the effectiveness of $C + 1$ classifier on semantic segmentation, and then transfer to object detection with some minor adjustments. During inference we use the output of softmax layer for classification.

### 4.1 SETTINGS

**Semantic Segmentation**. Semantic segmentation aims to label each pixel of an image with one semantic class label or background label. We evaluate our methods on two popular semantic segmentation dataset: PASCAL VOC and PASCAL Context and a human parsing dataset LIP. To make a fair comparison, for PASCAL VOC 2012 and PASCAL Context we adopt MMSegmentation Contributors (2020) as a unified framework for the experiments on semantic segmentation.

PASCAL VOC 2012 contains 20 foreground object classes and one background class. The original dataset contains 1,464 (train), 1,449 (val), and 1,456 (test) pixel-level annotated images. Following the settings in Chen et al. (2018b), we use the augmented dataset by the extra annotations provided by Everingham et al. (2015), which contains 10,582 (trainaug) training images. Following the setting in MMSegmmentation Contributors (2020), we resize the images into $512 \times 512$ and the output stride is 8. We adopt "SGD" as the optimizer and "poly" policy as the learning rate schedule. In addition, we set the initial learning rate as 0.01 and weight decay as 0.0005. Furthermore, the batch size is 16 and the number of iterations is 20K. We evaluate the performance of our method and other methods using single scale and without flipping.

PASCAL Context Mottaghi et al. (2014) contains 459 labeled categories, including 10,103 images, of which 4,998 are used for training and 5,105 for validation. The most widely adopted setting is to use the most frequent 59 categories as the semantic object classes and all the remaining categories as background. Following the setting in MMSegmmentation, we resize the images to $480 \times 480$. The initial learning rate is set to 0.004 and weight decay to 0.0001. The batch size is 16 and the number of iterations is 40K. We evaluate the performance using single scale and without flipping.

LIP Gong et al. (2017) is a human part segmentation dataset, with 50,462 images in total, including 30,462 images for training, 10,000 images for validation and 10,000 images for testing. In addition, it contains 19 semantic classes and 1 background class. We resize the images to $473 \times 473$. The initial learning rate is set to 0.0028 and weight decay to 0.0005. The batch size is 16 and the number of iterations is 110K. We evaluate the performance using single scale and flipping.

Table 2: Effect of three different center representations

| Center representation | mIoU | mAcc | aAcc |
|---|---|---|---|
| Learnable | 76.68 | 86.29 | 94.71 |
| Moving average | **77.35** | **86.70** | **94.87** |
| Shared | 76.99 | 86.57 | 94.79 |

### 4.2 SEMANTIC SEGMENTATION

#### 4.2.1 ABLATION STUDY

For ablation study, we use VOC+Aug as the training set. All images are resized to 512x512 as input. Iteration number is 20K. The initial learning rate is 0.01, and then decayed by "poly" policy with

Table 3: Weights' effect of $L_{compact}$

| Weights for $L_{compact}$ | mIoU | mAcc | aAcc |
|---|---|---|---|
| 0.01 | 76.82 | 86.43 | 94.63 |
| 0.1 | **77.35** | **86.70** | **94.87** |
| 1 | 75.22 | 85.25 | 94.39 |

Table 4: Weights' effect of $L_{background}$

| Weights for $L_{background}$ | mIoU | mAcc | aAcc |
|---|---|---|---|
| 0.001 | 75.93 | 86.04 | 94.51 |
| 0.0001 | **77.35** | **86.70** | **94.87** |
| 0.00001 | 76.47 | 85.91 | 94.71 |

power being 0.9. Batch size is set to 16. We use the same setting for the all ablation study unless specified otherwise.

At first, we validate the effectiveness of $C + 1$ classifier on PASCAL VOC by DeeplabV3+ Chen et al. (2018a) with ResNet50 He et al. (2016) as backbone. As for the $C$ CoIs, we use moving averages of $L_2$-normalized features as centers. We set the weights of $L_{compact}$ and $L_{background}$ as 0.1 and 0.0001 respectively if each of them is used.

From the experiment performance on table 1, we observe that $L_{compact}$ and $L_{background}$ can both boost the performance significantly. When $L_{compact}$ is used with $L_{separability}$, we can improve mIoU by 0.32. If $L_{background}$ added, we can get positive improvement 1.10 furthermore, and $C + 1$ loss in total can improve the mIoU by 1.42. For semantic segmentation task, this is a significant improvement.

Then, we validate the effect of three different center representations on the performance. In this group experiments, we set the weighs of $L_{compact}$ and $L_{background}$ as 0.1 and 0.0001 respectively. We use the same hyper-parameters as above. From the experiment performance on table 2, we could find that moving average achieves the best performance.

Next we analyze the effect of weights of $L_{compact}$ and $L_{background}$. According to the above experiments, moving average is the best center representation, so we adopt moving average in this and following experiments. First, we observe the weights' effect of $L_{compact}$ while set the weight of $L_{background}$ as 0.0001. From table 3 we can observe that 0.1 is a reasonable weight for $L_{compact}$. Then we test the weight's effect of $L_{background}$ while set the weight of $L_{compact}$ as 0.1. From table 4 we find that 0.0001 is a reasonable weight for $L_{background}$. From experiment performance, we can conclude that setting the weights of $L_{compact}$ and $L_{background}$ as 0.1 and 0.0001 respectively is a proper choice.

Finally, we list the performance details of all classes on PASCAL VOC with and without $C + 1$ loss in table 5. We can observe our method get superior performance over baseline on 13 classes which are highlighed by boldface.

### 4.2.2 BACKBONE

To prove the generality of $C + 1$ classifier, we experiment on PASCAL VOC by DeepLabV3+ with ResNet101 as the backbone, displayed in table 6. Considering the larger capacity of ResNet101, we set the iteration as 40K and other hyper-parameters the same as with ResNet50 as the backbone. We can observe that $C+1$ classifier is also better than the baseline. Moving average is used as the center representation. The weights of $L_{compact}$ and $L_{background}$ are set to 0.01 and 0.0001 respectively.

### 4.2.3 COMPARISON WITH SOTA

To prove the generality of $C + 1$ classifier, we further experiment on PASCAL VOC by other SOTA semantic segmentation algorithms, i.e. HRNet Sun et al. (2019) and OCRNet Yuan et al. (2020). Both of them take HRNetW48 as backbone.We set the hyper-parameters the same as DeeplabV3+

Table 5: The performance details of all classes on PASCAL VOC with and without $C + 1$ loss

| Category | $L_{separability}$ | | $L_{C+1}$ | | Category | $L_{separability}$ | | $L_{C+1}$ | |
|---|---|---|---|---|---|---|---|---|---|
| | mIoU | mAcc | mIoU | mAcc | | mIoU | mAcc | mIoU | mAcc |
| aeroplane | 90.13 | 96.01 | **92.03** | 95.38 | diningtable | 57.14 | 60.24 | **54.87** | 57.38 |
| bicycle | 41.75 | 91.74 | 41.56 | 95.38 | dog | 84.18 | 93.75 | **86.26** | 94.38 |
| bird | 86.62 | 92.32 | **87.71** | 94.17 | horse | 84.86 | 88.88 | 84.74 | 92.98 |
| boat | 70.92 | 87.90 | **72.43** | 87.27 | motorbike | 84.51 | 91.09 | 83.90 | 92.30 |
| bottle | 76.63 | 87.79 | **78.37** | 89.66 | person | 84.99 | 91.61 | 85.08 | 91.08 |
| bus | 94.68 | 97.49 | **95.05** | 96.99 | pottedplant | 60.82 | 70.29 | 59.70 | 70.70 |
| car | 86.40 | 93.06 | 86.03 | 93.85 | sheep | 83.93 | 92.37 | **88.47** | 91.98 |
| cat | 90.50 | 94.84 | **93.10** | 96.94 | sofa | 46.29 | 53.87 | **55.80** | 65.52 |
| chair | 36.36 | 55.11 | **37.44** | 54.77 | train | 90.21 | 94.10 | 89.40 | 93.97 |
| cow | 86.65 | 90.30 | **87.68** | 90.18 | tv/monitor | 62.69 | 75.31 | **70.45** | 80.79 |
| background | 94.21 | 97.33 | 94.35 | 97.32 | | | | | |

Table 6: Experiment results on PASCAL VOC by DeeplabV3+ with ResNet101 as the backbone

| Loss | mIoU | mAcc | aAcc |
|---|---|---|---|
| DeepLabV3+ Chen et al. (2018a) | 78.62 | 86.55 | 95.22 |
| DeepLabV3++ours | **80.00** | **87.99** | **95.50** |

with ResNet101 as the backbone, including center representation and iteration. For OCRNet+ours, we set the weights of $L_{compact}$ and $L_{background}$ are set to 0.1 and 0.0001 respectively, and HRNet+FCN+ours with 0.01 and 0.0001 respectively. Results in table 7 shows that our C+1 classifier is also applicable to other semantic segmentation algorithms. Especially on the OCRNet, our method can improve the baseline with 0.99 mIoU.

### 4.2.4 EXPERIMENTS ON PASCAL CONTEXT

To prove the generalization ability of our classifier to other scenarios, we also experiment DeeplabV3+ with ResNet50 as backbone on other semantic segmentation dataset, i.e. PASCAL Context. We use the moving average as the center representation. All images are resized to 480x480. Iteration number is 40K. Initial learning rate is set to 0.004. Batch size is set to 16. From table 8, we can observe our classifier can also improve the performance on PASCAL Context too.

### 4.3 HUMAN PARSING

For LIP, all images are resized to 473x473. The initial learning rate and weight decay are set to 0.0028 and 0.0005 respectively. The batch size is 16 and the number of iterations is 110K. Experimental results are shown in Table 9.

There are also some other datasets, i.e. Cityscapes Cordts et al. (2016), KITTI Geiger et al. (2013) and ADE20K Zhou et al. (2017; 2019). However these datasets almost have no background annotations, so it cannot be defined as a $C + 1$ classification problem. Thus we don't experiment on them.

Table 7: Experiment results on different SOTA semantic segmentation algorithms

| Method | mIoU | mAcc | aAcc |
|---|---|---|---|
| HRNet48+FCN Sun et al. (2019) | 76.23 | 84.95 | 94.66 |
| HRNet48+FCN+Ours | **76.96** | **85.38** | **94.81** |
| OCRNet Yuan et al. (2020) | 77.14 | 85.92 | 94.94 |
| OCRNet+ours | **78.13** | **86.92** | **95.09** |

Table 8: Experiment results on PASCAL Context

| Method | mIoU | mAcc | aAcc |
|---|---|---|---|
| DeepLabV3+ Chen et al. (2018a) | 47.34 | 57.40 | 74.18 |
| DeepLabV3++ours | **47.81** | **58.29** | **74.68** |

Table 9: Experiment results on LIP

| Method | Flip-test | mIoU | mAcc | aAcc |
|---|---|---|---|---|
| HRNet Sun et al. (2019) | N | 53.42 | 65.20 | 86.80 |
| | Y | 54.53 | 65.77 | 87.30 |
| HRNet+ours | N | **53.76** | **65.63** | **86.81** |
| | Y | **54.78** | **66.19** | **87.32** |

## 4.4 OBJECT DETECTION

**Object detection**. In this section, we migrated our approach to the object detection task. Nowadays, existing methods for object detection can be divided into anchor-based and anchor-free according to whether anchors are needed. Among them, the anchor-based methods are quite different from the segmentation task in the classification head. Specifically, semantic segmentation is the classification of pixels. The target to be classified corresponds to the feature map position one-by-one, while the anchor-based methods needs to classify the anchor boxes, which is multiple to the feature map position. Therefore, we choose to verify our approach on the FCOS Tian et al. (2019) model which is the most similar method in object detection to the FCN-based segmentation network.

To make a fair comparison, we adopted MMDetection Chen et al. (2019b) as a unified framework for all the experiments on object detection. We use COOC2017 as the evaluation dataset for object detection. COOC2017 contains 80 annotated targets, with 118K training images, 5K validation images and 20K testing images. The standard COO-style evaluation is adopted. To compare fairly, we use the public MMDetection platform with the provided training setup and learning rate schedule for 2x. The batch size is set as 16.

The experimental results in table 10 show that our method can improve the detection performance of small objects.

## 5 CONCLUSION

In this paper, firstly we define the $C + 1$ classification problem. Then we propose a uniform abstract $C + 1$ loss for training the $C + 1$ classifier. Furthermore we design a instantialized $C + 1$ loss and prove its superiority over popular cross-entropy loss on some CV tasks, including semantic segmentation and object detection. In the future, we will explore more instantialized object for the $C + 1$ loss and experiment them on more problems, proving its generalizability on common $C + 1$ classification.

Table 10: Experiment results on COCO

| Method | AP | AP$_{50}$ | $AP_{75}$ | AP$_S$ | AP$_M$ | AP$_L$ |
|---|---|---|---|---|---|---|
| FCOS(R50) Tian et al. (2019) | 38.5 | 57.7 | 41.0 | 21.9 | 42.8 | 48.6 |
| FCOS(R50)+ours | **39.0** | **58.5** | **41.6** | **23.1** | **43.0** | **49.5** |
| FCOS(X101-64x4d) Tian et al. (2019) | 42.6 | 62.3 | 45.6 | 25.7 | 46.3 | 54.6 |
| FCOS(X101-64x4d)+ours | **43.0** | **62.9** | **46.1** | **27.2** | **46.8** | **54.5** |

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

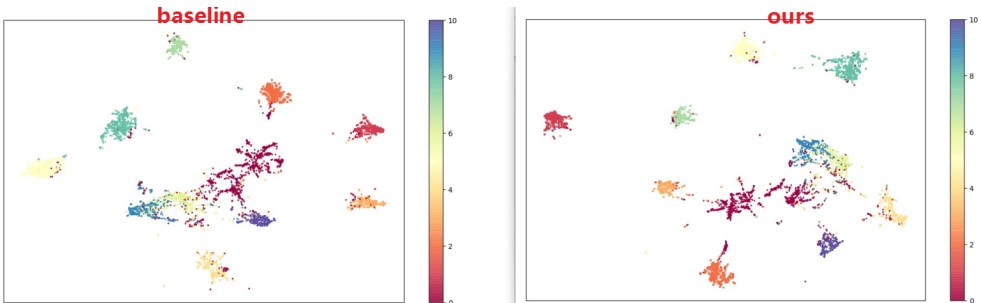

Figure 2: Feature visualization for 10 foreground classes and the background class in PASCAL VOC. 0 represents background class, 1 9 represent the selected 10 CoIs. View best in color.

## A  FEATURE VISUALIZATION

We visualized the features of 10 foreground classes and the background class in PASCAL VOC as displayed in Figure 2. The features are extracted by the DeepLabV3++ours method. We take each pixel of the feature map before classification layer as a feature vector. By adopting our method, the feature space of every CoI became more compact, and further away from background class. That is, there is large enough margin between background class and every CoI.

