# OpenReview forum: "C+1 Loss: Learn to Classify C Classes of Interest and the Background Class Differentially"
_ICLR.cc/2022/Conference — ICLR 2022 Submitted_

### Official Review · Reviewer_dnQ2 · 2021-11-01

**Correctness:** 2
**Technical Novelty And Significance:** 1
**Empirical Novelty And Significance:** 1
**Recommendation:** 3
**Confidence:** 4

**Main Review:**

Strengths:
+ The employed additional loss functions improved the original methods without these additional loss terms.

Weaknesses:
- Lack of novelty. The C+1 classification is a straightforward and widely adopted setting, which is the default for segmentation task, while the paper claims that "We define the C + 1 classification problem ..."; The method itself is basically adding two additional regularization terms using the center loss, which is employed from existing methods.
- With the claim being handling C+1 classification problem, methods from the OSR literature should be compared.
- Though an improvement on overall classification performance is shown, however, no evidence is provided showing that the representation is more compact and the margin has been full filled.
- No evidence is shown to support the conclusion of equation (3). The radii of the small and large balls should be demonstrated (e.g., by feature visualization).
- The writing is unsatisfactory, with grammatical errors and typos even in the abstract. For example, CoI is never mentioned before the short form appears; "In spite of ... use" ->  "In spite of ... using". The main context contains more issues...

**Summary Of The Paper:**

This paper aims at dealing with the C+1 classification problem, where C foreground classes and a background class, which could consist of other classes that do not overlap with the C foreground classes. The paper additionally includes center loss for foreground classes and a modified center loss for background class(es), on top of the standard cross-entropy loss. Results are reported on PASCAL VOC, PASCAL Context and LIP datasets.

**Summary Of The Review:**

Overall, the paper severely lacks novelty. The method simply adopts a loss function introduced in the existing method and a slightly modified version of it to improve the performance. The claims are not well supported by the experiments. The writing is unsatisfactory.

---

> ### Author Response · Authors · 2021-11-21
> **We are the first to propose to lean foreground classes and background class differentiably.**
>
> Indeed the C+1 classification is a widely adopted setting, but all of them treat all classes without difference. As far as we know, we are the first to differentiate the background class from foreground classes. Almost all algorithms of segmentation and detection adopt softmax loss or its variants (such as Focal Loss) as the default loss for classification, which learn the foreground and background with no difference.
> Does OSR mean Open-Set Recognition? If it does, we think we don’t need to compare with it because of their difference. In OSR, there aren’t any samples for the background class during training, but the background class should be detected during testing. However, the C+1 classification problem include many training samples for background class. As far as we know, there are no methods which adopt OSR algorithms as their classifier in the traditional semantic segmentation and object detection tasks, i.e PASCAL VOC, PASCAL Context, LIP datasets, and COCO.
> We have added a feature visualization on appendix of the newly submitted paper.

---

> > ### Comment · Reviewer_dnQ2 · 2021-11-29
> > **Keep original rating**
> >
> > I appreciate the authors' response. However, as agreed by the majority of reviewers, the paper lacks novelty and the quality needs significant improvement. The response did not add extra evidence to address these concerns. Thus, I would like to keep my original rating.
> > I mention Open-Set-Recognition(OSR) because OSR also considers the open-set classes, which corresponds to the C+1th class in the context of this paper, while OSR methods can achieve good results even without training with any open-set/background classes. Hence, better performance would be expected if the background training data is used, which should be validated experimentally. However, it could also be the case that adding the extra background training data leads to worse performance because of overfitting to the trained background. In this case, it is questionable to add the extra background training data. Unfortunately, this is not validated in the paper.

---

### Official Review · Reviewer_ZXUD · 2021-11-02

**Correctness:** 4
**Technical Novelty And Significance:** 2
**Empirical Novelty And Significance:** 3
**Recommendation:** 6
**Confidence:** 4

**Main Review:**

* Significance. The problem addressed by the author is important (the background class is present in most of the vision benchmarks) but few work (to my knowledge) attempted to tackle it in a rigorous way.

* Novelty. The solution proposed by the authors (decompose the loss function into three terms, to enforce some constraints of the last layer feature space) is original, easy to implement and has the potential to inspire others. The properties identified by the authors (which translate into these loss terms) are well discussed and intuitive.
 It might however be surprising  to maintain the basic loss (cross entropy preceded by a softmax) as such (since it probably acts against the compactness and background margin properties enforced by the two other terms).

* Results. The authors report convincing results on four different datasets and two different tasks. The experiments (ablation study) validate the claims of the introduction. While the quantitative analysis consistently show improvement wrt to baseline or the sota, I believe that more challenging experiments could have
been performed for binary classification tasks (eg, human detection/classification, road sign detection), for which the background is often very diverse.

* Clarity. The paper is fluid, a pleasure to read.

**Summary Of The Paper:**

The paper addresses the problem of semantic segmentation/classification, in situations where one has C+1 classes at hand, including  C deterministic classes and a background class. The authors identifies the weakness of the classical softmax classification layer in this context, and propose a novel loss, that can better account for the particularity of the non-deterministic background class. Experiments are performed on VOC-2012, Pascal context, LIP and COCO2017. Quantitative results, ablation study and comparison with related work are reported.


**Summary Of The Review:**

I enjoyed reading the paper because it tackles a well identified problem with clarity and concision.
The problem is well defined, the solution is simple and explained with clarity.
I believe that it belongs to the category of papers that I read once, keep it in memory, and come back to it regularly.

---------------------------------------------------
Updates: Thanks for the authors' response, which partially addresses my concerns.  I appreciate the fact that the authors specifically point out in their paper that the background class should be treated in a differentiated way wrt other classes. This observation, in itself, in my view, constitutes the originality of the work --even though the technical solution is surprisingly simple. I encourage the authors to re-submit to an other venue.

---

> ### Author Response · Authors · 2021-11-21
> **The three loss terms have a progressive enhancing effect while adding them one by one.**
>
> Maybe it’s more appropriate to call the first term as separability loss, which makes all classes separable. The compactness drives the representation space of C classes of interest more compact which is beneficial for the separability of C classes of interest. The background margin drives the representation space of background far away from the C classes of interest, which enhance the separability of background and C classes of interest. So the three loss terms are mutually beneficial, not conflicting.
> In this paper, we focus on validating our idea on some general C+1 classification tasks. Indeed, binary classification tasks are interesting. And we will consider some explorations on them in future.

---

### Official Review · Reviewer_NYF4 · 2021-11-02

**Correctness:** 3
**Technical Novelty And Significance:** 1
**Empirical Novelty And Significance:** 1
**Recommendation:** 1
**Confidence:** 5

**Main Review:**

The paper tackles an interesting and fair aspect in the field of multiclass classification. However, the neither the idea nor the kind of solution are new. In addition, the experimental results are not promising and show just slightly different results compared to arbitrarily selected baselines. From this points of view, there is neither a novel contribution nor provides the paper thrilling new insights to the tackled problem. In addition, the paper is not written and structured very well. For instance, the mathematical writing and the bibliography need to be seriously checked!

**Summary Of The Paper:**

The paper on hands tackles the problem of multi-class classification in the presence of a general  background class. This is illustrated by a specific loss function covering this aspect and demonstrated for a well known, but very old benchmark,

**Summary Of The Review:**

To summarize, the paper covers an interesting aspect, however, the overall novelty of the paper is very slim and the quality needs to be seriously improved. Thus, there is the clear recommendation not to accept the paper for ICLR!

---

> ### Author Response · Authors · 2021-11-21
> **We have compared our method with SOTA, such as OCRNet.**
>
> OCRNet and FCOS are the SOTA methods of semantic segmentation and object detection respectively, not the arbitrarily selected basleines. We select a representative method DeepLabV3+ as baseline for ablation study, and then transfer our method to different algorithms on different semantic segmentation dataset. Furthermore, we have validate our method on different tasks, including human parsing and object detection. All of this proved the generality of our method.

---

### Official Review · Reviewer_74g9 · 2021-11-03

**Correctness:** 3
**Technical Novelty And Significance:** 1
**Empirical Novelty And Significance:** 2
**Recommendation:** 3
**Confidence:** 5

**Main Review:**

Strengths
- The introduced C+1 loss is widely applicable to many kinds of tasks that require classification.
- The experiments are done extensively on multiple tasks to demonstrate the effectiveness of C+1 loss: object detection, semantic segmentation, human parsing.

Weaknesses
- C+1 loss is a simple and straightforward combination of existing loss functions already explored in previous papers `Wen et al. (2016)`. The background margin loss is almost identical to the conventional margin ranking loss. If you take things that are working really well and combine them in a simple way, it is likely that the combination will work better than any of the individual things.
- The naming of "basic discriminability" is so arbitrary and not supported by anything. Why does it have to be "basic"?
- When the paper mentions it uses center loss for L_compact, it does not cite the any papers about center loss.
- The proposed properties of C+1 loss are not backed by any theoretical foundations or insights.
- Except for DeepLabV3 on Pascal VOC, the performance gains are not so significant (most are less than 1%) in the evaluated tasks.


**Summary Of The Paper:**

This work is motivated by the different nature of the "other"/background class in many vision tasks such as object detection and semantic segmentation. It introduces C+1 loss which consists of 3 individual loss terms that focus on the basic discriminability, intra-class compactness, and background margin. Basic discriminability loss is the conventional classification loss for C+1 classes, intra-class compactness uses center loss, and background margin loss is a margin loss acted on the center representation of all C classes and the features of background samples.

**Summary Of The Review:**

This paper absolutely feels like a simple repackaging of old methods with a new name (C+1 loss) and there are almost no good enough contributions to make a case for acceptance.

---

> ### Author Response · Authors · 2021-11-21
> **Our work focus on validating our method on the widely-used benchmarks.**
>
> Indeed, “basic discriminability” is not the most appropriate term, we consider replacing it with “separability”.  Now we have cited the paper about center loss in the new submitted draft. We have revealed the necessity of the proposed properties. For OCRNet+ours on PASCAL VOC, the performance gain is about 1%. For FCOS(R50)+ours and FCOS(X101-64x4d)+ours on COCO, the performance gains are 1.2 and 1.5 for small object respectively. And small object detection is a challenging problem. Most importantly, our method doesn’t bring any more cost for inference.

---

### Author Response · Authors · 2021-11-21
**General response**

As far as we know, we are the first to propose the C+1 classification problem present all around the machine learning area. And we propose to learn the C classes of interest and background class differentially by a uniform abstract C+1 loss. These are the main contributions and innovation points of us. However, the instantialized C+1 loss is just one instance of the abstract C+1 loss, we could design any new loss accorded with our proposed properties. We have added a feature visualization on appendix of the newly submitted paper.

---

### Decision · Program_Chairs · 2022-01-20

**Decision:**

Reject

**Comment:**

This paper presents work on classification with a background class.  The reviewers appreciated the important, standard problem the paper considers.  However, concerns were raised regarding presentation, empirical evaluation, clarity, novelty, and signficance of the work.  The reviewers considered the authors' response in their subsequent discussions but felt the concerns were not adequately addressed.  Based on this feedback the paper is not yet ready for publication in ICLR.